# Retrospective study of more than 5 million emergency admissions to hospitals in England: Epidemiology and outcomes for people with dementia

**David Reeves**[1,2]*, **Fiona Holland**[1,2], **Hazel Morbey**[3], **Mark Hann**[1,2], **Faraz Ahmed**[3], **Linda Davies**[1], **John Keady**[4,5], **Iracema Leroi**[6], **Siobhan Reilly**[7]

1 National Institute for Health Research School for Primary Care Research, School of Health Sciences, Faculty of Biology, Medicine and Health, The University of Manchester, Manchester, United Kingdom, 2 Centre for Biostatistics, School of Health Sciences, Faculty of Biology, Medicine and Health, The University of Manchester, Manchester, United Kingdom, 3 Division of Health Research, Faculty of Health and Medicine, Lancaster University, Lancaster, United Kingdom, 4 National Institute for Health Research School for Social Care Research, Division of Nursing, Midwifery & Social Work, University of Manchester, Manchester, United Kingdom, 5 Greater Manchester Mental Health NHS Foundation Trust, Manchester, United Kingdom, 6 Department of Psychiatry St James' Hospital, Global Brain Health Institute, Trinity College Dublin, Dublin, Ireland, 7 Centre for Applied Dementia Studies, Faculty of Health Studies, University of Bradford, Bradford, United Kingdom

* david.reeves@manchester.ac.uk

**Data Availability Statement:** Electronic health records are, by definition, considered sensitive data in the UK by the Data Protection Act and cannot be

## Abstract

### Introduction

People living with dementia (PwD) admitted in emergency to an acute hospital may be at higher risk of inappropriate care and poorer outcomes including longer hospitalisations and higher risk of emergency re-admission or death. Since 2009 numerous national and local initiatives in England have sought to improve hospital care for PwD. We compared outcomes of emergency admissions for cohorts of patients aged 65+ with and without dementia at three points in time.

### Methods

We analysed emergency admissions (EAs) from the Hospital Episodes Statistics datasets for England 2010/11, 2012/13 and 2016/17. Dementia upon admission was based on a diagnosis in the patient's hospital records within the last five years. Outcomes were length of hospital stays (LoS), long stays (> = 15 days), emergency re-admissions (ERAs) and death in hospital or within 30 days post-discharge. A wide range of covariates were taken into account, including patient demographics, pre-existing health and reasons for admission. Hierarchical multivariable regression analysis, applied separately for males and females, estimated group differences adjusted for covariates.

### Results

We included 178 acute hospitals and 5,580,106 EAs, of which 356,992 (13.9%) were male PwD and 561,349 (18.6%) female PwD. Uncontrolled differences in outcomes between the

shared via public deposition because of information governance restrictions in place to protect patient confidentiality. For researchers who meet the criteria for access to confidential data, the Hospital Episode Statistics and linked Office of National Statistics mortality data used for this study is available via application to the UK Health and Social Care Information Centre. For details see: https://digital.nhs.uk/data-and-information/data-tools-and-services/data-services/hospital-episode-statistics.

**Funding:** This study was funded jointly by the Economic and Social Research Council (ESRC) and the National Institute for Health Research (NIHR). ESRC Grant reference: ES/L001772/1. The views expressed are those of the author(s) and not necessarily those of the ESRC, UKRI, NHS, the NIHR or the Department of Health and Social Care. This work forms part of the ESRC/NIHR Neighbourhoods and Dementia: a mixed methods study [https://sites.manchester.ac.uk/neighbourhoods-and-dementia/]. This paper relates to work programme 5: Developing the evidence base for dementia training in acute NHS hospitals (DEMTRAIN). The funding bodies had no role in the study design, data collection and analysis, decision to publish, or preparation of the manuscript.

**Competing interests:** The authors have declared that no competing interests exist.

patient groups were substantial but were considerably reduced after control for covariates. Covariate-adjusted differences in LoS were similar at all time-points and in 2016/17 were 17% (95%CI 15%-18%) and 12% (10%-14%) longer for male and female PwD respectively compared to patients without dementia. Adjusted excess risk of an ERA for PwD reduced over time to 17% (15%-18%) for males and 17% (16%-19%) for females, but principally due to increased ERA rates amongst patients without dementia. Adjusted overall mortality was 30% to 40% higher for PwD of both sexes throughout the time-period; however, adjusted in-hospital rates of mortality differed only slightly between the patient groups, whereas PwD had around double the risk of dying within 30 days of being discharged.

## Conclusion

Over the six-year period, covariate-adjusted hospital LoS, ERA rates and in-hospital mortality rates for PwD were only slightly elevated compared to similar patients without dementia and remaining differences potentially reflect uncontrolled confounding. PwD however, were around twice as likely to die shortly after discharge, the reasons for which require further investigation. Despite being widely used for service evaluation, LoS, ERA and mortality may lack sensitivity to changes in hospital care and support to PwD.

## Introduction

The number of people with dementia (PwD) in the United Kingdom (UK) is forecast to nearly double between 2021 and 2051 to around 1.7 million (p23) [1]. People with dementia are more likely to experience a hospitalisation [2, 3] and estimates suggest that at any one time up to 25%, or possibly more, of beds in UK hospitals are occupied by someone with dementia [4, 5]. Numerous studies have reported hospital in-patients with dementia to be at higher risk of inappropriate care and poorer outcomes, including poorer inter-personal care [6], greater risk of adverse events and complications (e.g. a fall, dehydration, bed sores, infections) [4, 7, 8], longer hospitalisations and higher risk of death or of emergency re-admission [7, 9, 10].

Against this background the first national strategy for dementia in England was introduced in 2009, with improved quality of care in general hospitals a key objective [11]. This was followed by a series of related initiatives, including the first national audit of dementia (NAD) care in general hospitals in 2010, financial incentivisation of screening upon admission for memory problems in 2012 through the dementia CQUIN (Commissioning for Quality and Innovation) payment [12], the introduction in 2013 by Health Education England of progressive increases in provision of foundation-level dementia training for all staff looking after patients with dementia [13, 14], publication of the Five Year Forward View vision for the NHS [15] and Dementia Roadmap web-resource [16] in 2014, and publication of the Dementia Core Skills Education and Training Framework in 2015 (updated 2018) [17] detailing the essential skills and knowledge necessary across health and social care. Numerous other local and national initiatives to improve care, such as the Butterfly Scheme [18] and John's Campaign [19] have also become widespread. Yet despite this considerable activity, evaluation has been lacking on whether disparities in hospital outcomes for people with and without dementia have reduced as a result.

An understanding of the true extent of the differences in outcomes for people with dementia compared to people without dementia (PwoD) and the factors associated, is important for

the design of effective innovations and interventions. Accurate comparisons are made complex because PwD differ from other in-patients in many ways: they are generally older [20] and may have a greater illness burden upon admission [21], they may also have different patterns of co-morbidities [21] and be admitted for different conditions [21–23]. Very few studies have controlled for such differences to any appreciable degree. Hence it is not clear to what extent the care received in hospital per se contributes to differences in observed outcomes, leading to calls for greater attention to be paid to this issue [10].

### Aims

We used the NHS-Digital Hospital Episode Statistics (HES) dataset for England [24] and analysed emergency admissions for physical health conditions. Our aims were:

1. To describe the epidemiology of emergency hospital admissions for PwD and how this compares to PwoD, across the period 2010 to 2016.

2. To examine the trends in hospital outcomes for PwD and PwoD across the period and the extent to which differences in outcomes might be accounted for by differences in patient characteristics.

## Methods

### Source

This study is based on the English National Health Service (NHS) Hospital Episode Statistics (HES) database, linked to Office of National Statistics (ONS) official mortality data and pseudonymised for the purposes of research. The data was obtained via application to the NHS Digital Data Access Request Service (DARS) approved on 6/4/2017, DARS reference NIC 33318. Specifically, we obtained extracts of individual patient data from the admitted patient care (APC) dataset within HES for the financial years (FYs; UK government accounting years, which run from April 1st to the following March 31st) 2010/11, 2012/13 and 2016/17. The APC dataset contains detailed data on all admissions to NHS hospitals in England [24]. Professional coders enter the data retrospectively using information collected from the medical records. Up to 20 diagnosis fields are used to capture the primary diagnosis for each admission plus other relevant diagnoses, using ICD-10 codes. We obtained diagnosis data for every FY from 2005/6 to 2016/17. Following the method of previous studies [25], for each admission the patient was classified as having comorbid dementia if their record included an ICD10 dementia code as secondary diagnosis for that admission, or as a primary or secondary diagnosis for any previous admission at any hospital over the prior 5 FYs. For this purpose we adapted the set of ICD-10 codes used by the NAD survey in 2016/17 (See Box A1 in S1 File) [26]. Individuals coded as having dementia at any point in a FY were assumed to have that status for any other admission during that FY. We had no ability at any point to identify specific individuals within the datasets.

### Inclusion and exclusion criteria

**Hospitals.** Using the APC database we identified general acute hospitals and excluded other providers such as day, specialist, palliative and rehabilitation units. We also excluded a few atypically small providers and acute hospitals that did not contribute data to all three FYs 2010/11, 2012/13 and 2016/17.

**Patients.** For each hospital in each FY, we extracted data for people aged 65 or above with one or more emergency admissions (EA) for a physical health condition. A hospital "spell"

was defined as a continuous stay in hospital, made up of one or more finished consultant episodes (FCE) of care. Patients could have multiple spells in a FY, though spells that extended into the next FY were necessarily excluded due to data limits. We included spells coded as Eas and excluded elective admissions, day cases and regular day and night attenders. To focus on Eas for physical health conditions we excluded spells where a mental health condition, including dementia, was the main speciality or primary diagnosis, or for palliative care. Data analysis was undertaken between 2018 and 2022.

## Outcomes

Outcomes for patients were computed for each complete spell. Length of hospital stay (LoS) was calculated as the number of days between admission and subsequent discharge including all consultant episodes over that period. Long stays were spells of 15 days or more. An Emergency re-admission (ERA) was an admission for an emergency within 30 days of a previous discharge. Death was a death whilst in hospital or within 30 days of discharge. We analysed total deaths and also sub-groups of deaths in hospital and deaths within 30 days of discharge, with the latter sample restricted to individuals who had survived to be discharged. Finally we analysed ERA or Death as a combined outcome. Full details of each outcome are provided in Box A2 in S1 File.

## Clinical factors

Reasons for an EA varied by gender and a few medical procedures were gender-specific. We therefore conducted separate analyses for males and females. Using the APC dataset we constructed a wide range of clinical factors related to each hospital spell, including patient demographics and pre-existing health along with specifics of the spell itself. These were used to characterise and compare the patient cohorts and also as covariates in regression analyses. We used a fine-grained categorisation of each factor to capture as much association with patient group as possible, with missing values as an additional category to keep all spells in the analysis. The clinical factors were grouped into subsets as summarised below. Full details are provided in Box A2 in S1 File.

Demographic characteristics: age on admission (in 5-year age-bands); ethnicity; residential area Index of Multiple Deprivation (IMD) quintile; residential area low income quintile.

Spell characteristics: month of admission (in seasonal quarters); weekend admission; source of admission (usual residence, other hospital, residential care/care home, other/unknown).

Pre-existing health: comorbidities (30 Elixhauser comorbidities [27, 28]); number of Eas in the previous financial year.

Admitting condition: the primary reason for admission and type of treatment given (based on the first episode core Heath Resource Group (HRG) subchapter) [29].

Number of consultant episodes: number of specialist consultants providing care over the spell.

Discharge destination: The place discharged to (usual residence, other hospital, residential care/care home, died in hospital, other/unknown).

## Statistical models

Descriptive statistics were used to characterise and contrast the patient samples.

Patients and spells were clustered within hospitals. Multi-level models (MLMs) such as random effect, fixed effect and mixed-effect models attempt to model this structure and are often recommended for this type of data. This study involved a large number of separate analyses and MLMs frequently failed to converge to a solution due to very large samples and model

complexity. An alternative approach consists of analysing the data as a single-level model and utilising cluster-robust estimates of variance to account for inter-correlations between observations within clusters [30–32]. This approach makes fewer assumptions than MLMs and is appropriate when the focus is not on effects at the cluster level [31]. This method consistently converged and was used for our primary analyses. As a sensitivity, we also report results from MLMs for which convergence was achieved.

Length of stay demonstrated a highly skewed distribution so was treated within our models as a count variable with a negative binomial distribution to account for overdispersion. Long spells, ERAs, deaths, and combined ERA/death were analysed as binary variables using robust Poisson regression to obtain direct estimates of relative risks [33, 34].

Analyses focused on the extent to which clinical factors could potentially account for differences in outcomes between patients with and without dementia. We ran hierarchical regression models, with each model building on the previous by adding a further subset of covariates. The initial model (M1) included dementia status alone; M2 added the subset of demographic factors; M3 hospital spell characteristics; M4 measures of pre-existing health; M5 admitting condition (HRG subchapter); M6 number of consultant episodes; and M7 discharge destination. Due to the very high volume of data and model complexity, we analysed each FY as a separate cross-sectional dataset.

Number of consultant episodes (M6) and discharge destination (M7) can reflect the health care needs of a patient, but might also reflect events in hospital: for example, a hospital-acquired infection may result in an additional consultant episode. In this paper we make a distinction between these "post-admission" factors and the other "pre-admission" factors.

### Sensitivity analyses

Full details of the sensitivity analyses including rationales and statistical models are provided in the S1 File. Briefly, these included: exclusion of spells of <2 days; exclusion of spells with psychiatric comorbidities; analysis of a 1:1 matched sample of spells; use of MLMs (on a 50% random sample to increase convergence); re-analysis of LoS using a Fine and Gray survival model accounting for death in hospital as a competing risk; [23]; re-analysis of binary outcomes using logistic regression [35].

## Results

The sample consisted of 178 hospitals. Table 1 provides the numbers of eligible patients and Eas for each FY, broken down by dementia status and sex. The total sample consisted of 2,569,007 spells across 1,685,723 male patients, of which 356,992 spells (13.9%) involved PwD, and 3,011,099 spells across 2,028,743 female patients of which 561,349 spells (18.6%) involved PwD. Less than 6% of spells in any FY extended into the following FY and were thereby excluded (similar across patient groups).

### Sample descriptive statistics

Numbers of patients and spells increased year-on-year for PwD and PwoD of both sexes, though proportionately more for PwD (Table 1). The representation of male PwD increased from 9.8% of all male patients in 2010/11 to 13.3% in 2016/17, while the representation of female PwD increased from 14.8% to 17.7% of all female patients.

Descriptive statistics for the hospital spells were similar across the three FYs, so here we focus on FY 2016/17 (Table 2); for 2010/11 and 2012/13 see Tables A1 and A2 in S1 File. PwD of both sexes were much more likely to be older. The patient groups had similar large majorities with White ethnicity. They were also similar in residential IMD though PwD were slightly

**Table 1. Summary of analysis samples[a].**

| | Males | | | | | | Females | | | | | |
|---|---|---|---|---|---|---|---|---|---|---|---|---|
| | 2010/11 | | 2012/13 | | 2016/17 | | 2010/11 | | 2012/13 | | 2016/17 | |
| | PwD[b] | PwoD[c] | PwD | PwoD | PwD | PwoD | PwD | PwoD | PwD | PwoD | PwD | PwoD |
| **Patients** | | | | | | | | | | | | |
| Number of patients | 49,473 | 452,865 | 62,248 | 480,923 | 84,856 | 555,358 | 92,656 | 532,866 | 108,740 | 549,229 | 131,589 | 613,663 |
| % within the FY | 9.8 | 90.2 | 11.5 | 88.5 | 13.3 | 86.7 | 14.8 | 85.2 | 16.5 | 83.5 | 17.7 | 82.3 |
| **Emergency admissions** | | | | | | | | | | | | |
| Number of spells | 87,924 | 659,615 | 111,846 | 706,090 | 157,222 | 846,310 | 153,131 | 756,483 | 182,956 | 786,434 | 225,262 | 906,833 |
| % within the FY | 11.8 | 88.2 | 13.7 | 86.3 | 15.7 | 84.3 | 16.8 | 83.2 | 18.9 | 81.1 | 19.9 | 80.1 |

[a]Totals across 178 hospitals common to all three years

[b]PwD patients with dementia; [c]PwoD patients without dementia

more likely to live in areas in the two lowest income quintiles. Calendar months of admission were similar but PwD were a little more likely to be admitted on a weekend. The great majority of admissions were from a patient's "usual residence" with only small numbers being admitted from care homes. Most discharges were to a patient's usual residence, though PwD were more likely to be coded as discharged to a care home. Around 4.5% of both groups were discharged to another hospital.

Both male and female PwD were more likely to have experienced one or more EAs in the previous FY. Mean numbers of comorbidities were similar for both groups, though patients with dementia were more likely to have fluid and electrolyte disorders, additional neurodegenerative disorders and depression, but less likely to have cancer or be obese.

Regarding reasons for admission, most HRG subchapters were represented by less than 2% of patients in either group. Some conditions were more frequently encountered in PwD and others less frequently. This included higher rates of orthopaedic trauma and renal procedures/disorders, but lower rates of cardiac disorders and digestive system disorders. PwD were also more likely to have more than one consultant episode during their spell in hospital.

The demographics of the patient cohorts changed very little over the study period. Age and residential IMD distributions hardly differed between 2010/11 and 2016/17, though there was a slight decrease in the representation of white ethnic groups by around 2% to 3%.

### Descriptive statistics for outcomes

In all FYs mean LoS was longer for PwD of both sexes (Table 3; Fig 1A). Mean LoS declined slightly over time for both groups and in 2016/17 was 9.7 versus 6.2 days for males, and 9.6 versus 6.7 days for females. Rates of spells > = 15 days had a very similar pattern (Fig 1B). In all FYs PwD were more likely to experience an ERA and males more-so than females. Rates of ERAs were similar in 2010 and 2012, but increased a little in 2016/17, though slightly less for PwD (Fig 1C). By contrast, rates of deaths (in hospital or within 30 days of discharge) demonstrated a small reduction year-on-year for both patient groups and both sexes (Fig 1D), though remained substantially higher for PwD. The increase in ERAs and decrease in deaths essentially cancelled each other out in the combined outcome, leaving it stable over time, although higher throughout for PwD (Fig A1A in S1 File).

### Hierarchical regression analysis

Findings of the hierarchical regression analyses controlling for demographic, health and other differences between the patient groups are summarised in Table 4. Results–in the form of

**Table 2. Descriptive statistics for the FY2016/17 hospital spell sample.**

| | Males | | | | Females | | | |
|---|---|---|---|---|---|---|---|---|
| | PwD | | PwoD | | PwD | | PwoD | |
| | n | % | n | % | n | % | n | % |
| Total number of spells | 157,222 | 15.7 | 846,310 | 84.3 | 225,262 | 19.9 | 906,833 | 80.1 |
| **Patient demographics** | | | | | | | | |
| **Age on admission** | | | | | | | | |
| 65–69 | 7,678 | 4.9 | 169,864 | 20.1 | 5,639 | 2.5 | 146,453 | 16.1 |
| 70–74 | 14,383 | 9.1 | 173,295 | 20.5 | 12,251 | 5.4 | 155,706 | 17.2 |
| 75–79 | 26,287 | 16.7 | 169,595 | 20 | 26,633 | 11.8 | 167,804 | 18.5 |
| 80–84 | 39,959 | 25.4 | 157,220 | 18.6 | 48,444 | 21.5 | 173,062 | 19.1 |
| 85–89 | 41,290 | 26.3 | 112,019 | 13.2 | 65,877 | 29.2 | 149,195 | 16.5 |
| 90plus | 27,625 | 17.6 | 64,317 | 7.6 | 66,418 | 29.5 | 114,613 | 12.6 |
| Mean (SD) | 82.8 | 7.1 | 77.4 | 7.9 | 85.3 | 7.0 | 79.2 | 8.4 |
| **Ethnicity** | | | | | | | | |
| White | 138,301 | 88.0 | 742,366 | 87.7 | 201,304 | 89.4 | 800,216 | 88.2 |
| Black African | 556 | 0.4 | 2,394 | 0.3 | 463 | 0.2 | 2,502 | 0.3 |
| Black Caribbean | 2,223 | 1.4 | 6,504 | 0.8 | 2,215 | 1.0 | 6,665 | 0.7 |
| Indian | 1,969 | 1.3 | 13,003 | 1.5 | 2,038 | 0.9 | 13,240 | 1.5 |
| Pakistani | 1,242 | 0.8 | 7,415 | 0.9 | 1,059 | 0.5 | 7,493 | 0.8 |
| Bangladeshi | 455 | 0.3 | 2,099 | 0.2 | 343 | 0.2 | 1,859 | 0.2 |
| Black other/mixed | 546 | 0.3 | 2,162 | 0.3 | 532 | 0.2 | 2,180 | 0.2 |
| Asian other/mixed | 1,001 | 0.6 | 5,676 | 0.7 | 855 | 0.4 | 5,079 | 0.6 |
| Other ethnic group/mixed | 1,896 | 1.2 | 9,233 | 1.1 | 2,445 | 1.1 | 9,616 | 1.1 |
| Unknown | 9,033 | 5.7 | 55,458 | 6.6 | 14,008 | 6.2 | 57,983 | 6.4 |
| **Residential area IMD quintile (2015)** | | | | | | | | |
| 1 (most deprived) | 33,314 | 21.2 | 157,829 | 18.6 | 47,757 | 21.2 | 175,311 | 19.3 |
| 2 | 33,361 | 21.2 | 162,172 | 19.2 | 47,121 | 20.9 | 177,471 | 19.6 |
| 3 | 32,490 | 20.7 | 175,850 | 20.8 | 46,630 | 20.7 | 186,761 | 20.6 |
| 4 | 29,997 | 19.1 | 174,308 | 20.6 | 43,773 | 19.4 | 184,915 | 20.4 |
| 5 (least deprived) | 26,802 | 17 | 164,738 | 19.5 | 38,338 | 17.0 | 171,569 | 18.9 |
| Missing | 1,258 | 0.8 | 11,413 | 1.3 | 1,643 | 0.7 | 10,806 | 1.2 |
| **Residential area low income quintile** | | | | | | | | |
| 1 (Highest income) | 23,782 | 15.1 | 160,884 | 19 | 32,081 | 14.2 | 159,919 | 17.6 |
| 2 | 26,750 | 17 | 160,757 | 19 | 37,522 | 16.7 | 165,548 | 18.3 |
| 3 | 29,676 | 18.9 | 160,080 | 18.9 | 44,088 | 19.6 | 173,059 | 19.1 |
| 4 | 33,906 | 21.6 | 162,843 | 19.2 | 49,929 | 22.2 | 182,889 | 20.2 |
| 5 (Lowest income) | 37,662 | 24 | 170,290 | 20.1 | 54,198 | 24.1 | 192,804 | 21.3 |
| Missing | 5,446 | 3.5 | 31,456 | 3.7 | 7,444 | 3.3 | 32,614 | 3.6 |
| **Spell characteristics** | | | | | | | | |
| **Month of admission** | | | | | | | | |
| March-May | 37,213 | 23.7 | 203,777 | 24.1 | 52,927 | 23.5 | 216,490 | 23.9 |
| June-August | 40,444 | 25.7 | 212,012 | 25.1 | 57,540 | 25.5 | 226,831 | 25.0 |
| September-November | 39,609 | 25.2 | 210,016 | 24.8 | 56,596 | 25.1 | 225,064 | 24.8 |
| December-February | 39,956 | 25.4 | 220,505 | 26.1 | 58,199 | 25.8 | 238,448 | 26.3 |
| **Weekend admission (Sat or Sun)** | | | | | | | | |
| Yes | 42,850 | 27.3 | 198,640 | 23.5 | 63,061 | 28.0 | 217,259 | 24.0 |
| No | 114,372 | 72.7 | 647,670 | 76.5 | 162,201 | 72.0 | 689,574 | 76.0 |
| **Source of admission** | | | | | | | | |

*(Continued)*

**Table 2.** (Continued)

| | Males | | | | Females | | | |
|---|---|---|---|---|---|---|---|---|
| | PwD | | PwoD | | PwD | | PwoD | |
| | n | % | n | % | n | % | n | % |
| Usual residence | 144,838 | 92.1 | 803,496 | 94.9 | 207,217 | 92.0 | 863,261 | 95.2 |
| Other hospital | 7,508 | 4.8 | 37,159 | 4.4 | 9,726 | 4.3 | 36,694 | 4.0 |
| Residential care/care home | 4,334 | 2.8 | 2,723 | 0.3 | 7,597 | 3.4 | 3,892 | 0.4 |
| Other/unknown | 542 | 0.3 | 2,932 | 0.3 | 722 | 0.3 | 2,986 | 0.3 |
| **Pre-existing health** | | | | | | | | |
| **Elixhauser comorbidities** | | | | | | | | |
| Congestive heart failure | 39,417 | 25.1 | 202,291 | 23.9 | 51,346 | 22.8 | 196,205 | 21.6 |
| Cardiac arrhythmias | 76,040 | 48.4 | 353,101 | 41.7 | 91,329 | 40.5 | 323,307 | 35.7 |
| Valvular disease | 21,232 | 13.5 | 128,501 | 15.2 | 28,920 | 12.8 | 132,025 | 14.6 |
| Pulmonary circulation disorders | 6,328 | 4.0 | 49,814 | 5.9 | 10,010 | 4.4 | 59,111 | 6.5 |
| Peripheral vascular disorders | 20,842 | 13.3 | 120,149 | 14.2 | 15,813 | 7.0 | 71,935 | 7.9 |
| Hypertension | 100,879 | 64.2 | 548,018 | 64.8 | 148,327 | 65.8 | 595,216 | 65.6 |
| Paralysis | 9,420 | 6.0 | 32,274 | 3.8 | 9,750 | 4.3 | 28,989 | 3.2 |
| Other (non-dementia) neurodegenerative disorders | 42,281 | 26.9 | 84,884 | 10.0 | 41,195 | 18.3 | 78,402 | 8.6 |
| Chronic pulmonary disease | 45,256 | 28.8 | 275,439 | 32.5 | 57,954 | 25.7 | 296,598 | 32.7 |
| Diabetes, uncomplicated | 47,901 | 30.5 | 241,537 | 28.5 | 53,166 | 23.6 | 206,816 | 22.8 |
| Diabetes, complicated | 8,481 | 5.4 | 42,046 | 5.0 | 6,783 | 3.0 | 28,480 | 3.1 |
| Hypothyroidism | 12,104 | 7.7 | 50,595 | 6.0 | 40,355 | 17.9 | 151,341 | 16.7 |
| Renal failure | 47,385 | 30.1 | 199,247 | 23.5 | 61,871 | 27.5 | 191,114 | 21.1 |
| Liver disease | 5,837 | 3.7 | 43,384 | 5.1 | 5,538 | 2.5 | 36,790 | 4.1 |
| Peptic ulcer disease excluding bleeding | 3,993 | 2.5 | 23,668 | 2.8 | 3,155 | 1.4 | 18,250 | 2.0 |
| Lymphoma + AIDS/HIV | 1,681 | 1.1 | 20,301 | 2.4 | 1,603 | 0.7 | 15,512 | 1.7 |
| Metastatic cancer | 4,942 | 3.1 | 69,534 | 8.2 | 3,557 | 1.6 | 54,727 | 6.0 |
| Solid tumour without metastasis | 22,882 | 14.6 | 164,637 | 19.5 | 12,564 | 5.6 | 98,556 | 10.9 |
| Rheumatoid arthritis/collagen vascular diseases | 7,094 | 4.5 | 45,765 | 5.4 | 17,016 | 7.6 | 91,989 | 10.1 |
| Coagulopathy | 3,642 | 2.3 | 22,810 | 2.7 | 3,529 | 1.6 | 17,849 | 2.0 |
| Obesity | 5,631 | 3.6 | 58,297 | 6.9 | 7,432 | 3.3 | 63,969 | 7.1 |
| Weight loss | 8,307 | 5.3 | 45,116 | 5.3 | 9,734 | 4.3 | 43,413 | 4.8 |
| Fluid and electrolyte disorders | 59,493 | 37.8 | 205,671 | 24.3 | 85,691 | 38.0 | 236,530 | 26.1 |
| Blood loss anaemia | 742 | 0.5 | 3,939 | 0.5 | 765 | 0.3 | 3,431 | 0.4 |
| Deficiency anaemia | 18,675 | 11.9 | 86,761 | 10.3 | 28,776 | 12.8 | 102,924 | 11.3 |
| Alcohol abuse | 10,056 | 6.4 | 56,409 | 6.7 | 4,922 | 2.2 | 22,305 | 2.5 |
| Drug abuse | 308 | 0.2 | 2,289 | 0.3 | 416 | 0.2 | 1,353 | 0.1 |
| Psychoses | 3,679 | 2.3 | 7,557 | 0.9 | 5,665 | 2.5 | 9,456 | 1.0 |
| Depression | 22,030 | 14.0 | 63,272 | 7.5 | 40,353 | 17.9 | 103,427 | 11.4 |
| **Number of Elixhauser comorbidities** | | | | | | | | |
| Mean (SD) | 4.2 | 2.4 | 3.8 | 2.5 | 3.8 | 2.3 | 3.6 | 2.4 |
| **No. of emergency admissions in previous FY** | | | | | | | | |
| None | 71,714 | 45.6 | 555,413 | 65.6 | 110,567 | 49.1 | 597,795 | 65.9 |
| 1 | 36,825 | 23.4 | 150,124 | 17.7 | 54,316 | 24.1 | 165,213 | 18.2 |
| 2 | 20,423 | 13.0 | 65,443 | 7.7 | 27,982 | 12.4 | 69,956 | 7.7 |
| 3 | 11,255 | 7.2 | 31,656 | 3.7 | 14,139 | 6.3 | 33,082 | 3.6 |
| 4 | 6,486 | 4.1 | 16,433 | 1.9 | 7,795 | 3.5 | 16,927 | 1.9 |
| 5 | 3,628 | 2.3 | 9,855 | 1.2 | 4,143 | 1.8 | 9,425 | 1.0 |
| 6 | 2,265 | 1.4 | 5,432 | 0.6 | 2,272 | 1.0 | 5,065 | 0.6 |

(*Continued*)

**Table 2.** (Continued)

| | Males | | | | Females | | | |
|---|---|---|---|---|---|---|---|---|
| | PwD | | PwoD | | PwD | | PwoD | |
| | n | % | n | % | n | % | n | % |
| 7+ | 4,626 | 2.9 | 11,954 | 1.4 | 4,048 | 1.8 | 9,370 | 1.0 |
| **Admitting condition type and severity** | | | | | | | | |
| **Spell Core HRG subchapter** | | | | | | | | |
| AA+AB Nervous System Procedures and Disorders/Pain management | 11,082 | 7.0 | 54,298 | 6.4 | 15,682 | 7.0 | 58,462 | 6.4 |
| BZ Eyes and Periorbita Procedures and Disorders | 484 | 0.3 | 3,557 | 0.4 | 809 | 0.4 | 4,023 | 0.4 |
| CZ Mouth Head Neck and Ears Procedures and Disorders | 4,593 | 2.9 | 21,590 | 2.6 | 7,956 | 3.5 | 24,062 | 2.7 |
| DZ Respiratory System Procedures and Disorders | 32,857 | 20.9 | 150,957 | 17.8 | 40,678 | 18.1 | 157,863 | 17.4 |
| EA Cardiac Procedures | 1,282 | 0.8 | 28,628 | 3.4 | 925 | 0.4 | 17,212 | 1.9 |
| EB Cardiac Disorders | 14,213 | 9.0 | 116,120 | 13.7 | 18,981 | 8.4 | 118,334 | 13.0 |
| FZ Digestive System Procedures and Disorders | 10,913 | 6.9 | 86,077 | 10.2 | 16,420 | 7.3 | 101,275 | 11.2 |
| GA+GB Hepatobiliary and Pancreatic System Open, Laparoscopic, Endoscopic Procedures | 501 | 0.3 | 6,933 | 0.8 | 568 | 0.3 | 6,800 | 0.7 |
| GC Hepatobiliary and Pancreatic System Disorders | 1,372 | 0.9 | 16,810 | 2.0 | 1,893 | 0.8 | 16,572 | 1.8 |
| HA Orthopaedic Trauma Procedures | 7,374 | 4.7 | 26,107 | 3.1 | 21,098 | 9.4 | 59,261 | 6.5 |
| HB+HR Orthopaedic Non-Trauma + Reconstruction Procedures | 1,055 | 0.7 | 7,901 | 0.9 | 1,732 | 0.8 | 10,079 | 1.1 |
| HC Spinal Procedures and Disorders | 1,097 | 0.7 | 9,177 | 1.1 | 2,158 | 1.0 | 13,557 | 1.5 |
| HD Musculoskeletal and Rheumatological Disorders | 2,891 | 1.8 | 21,271 | 2.5 | 6,608 | 2.9 | 31,961 | 3.5 |
| JA+JB+JC Breast/Burn/Skin Procedures and Disorders | 651 | 0.4 | 6,111 | 0.7 | 1,174 | 0.5 | 7,097 | 0.8 |
| JD Skin Disorders | 2,591 | 1.6 | 18,477 | 2.2 | 4,583 | 2.0 | 21,230 | 2.3 |
| KA+KC Endocrine System/Metabolic Disorders | 1,790 | 1.1 | 9,919 | 1.2 | 3,563 | 1.6 | 13,899 | 1.5 |
| KB Diabetic Medicine | 1,599 | 1.0 | 5,808 | 0.7 | 2,060 | 0.9 | 4,883 | 0.5 |
| LA Renal Procedures and Disorders | 17,510 | 11.1 | 47,142 | 5.6 | 24,121 | 10.7 | 50,194 | 5.5 |
| LB Urological and Male Reproductive System Procedures and Disorders | 7,855 | 5.0 | 45,462 | 5.4 | 1,622 | 0.7 | 8,675 | 1.0 |
| MA+MB Female Reproductive System Procedures/Disorders | NA | NA | NA | NA | 463 | 0.2 | 3,697 | 0.4 |
| QZ Vascular Procedures and Disorders | 1,353 | 0.9 | 15,647 | 1.8 | 2,067 | 0.9 | 12,789 | 1.4 |
| SA Haematological Procedures and Disorders | 1,332 | 0.8 | 15,469 | 1.8 | 2,067 | 0.9 | 15,149 | 1.7 |
| VA Multiple Trauma | 1,759 | 1.1 | 6,738 | 0.8 | 3,820 | 1.7 | 10,837 | 1.2 |
| WA_A Immunology, Infectious Diseases, Poisoning | 4,472 | 2.8 | 19,873 | 2.3 | 5,584 | 2.5 | 19,807 | 2.2 |
| WA_B Complications of procedures | 331 | 0.2 | 5,953 | 0.7 | 288 | 0.1 | 5,003 | 0.6 |
| WA_D Unexplained symptoms or abnormal findings | 3,602 | 2.3 | 8,531 | 1.0 | 5,204 | 2.3 | 10,299 | 1.1 |
| RC+WA_C+WA_E+ZZ Miscellaneous | 8,562 | 5.4 | 22,231 | 2.6 | 13,398 | 5.9 | 25,631 | 2.8 |
| Missing or not applicable | 14,101 | 9.0 | 69,523 | 8.2 | 19,740 | 8.8 | 78,182 | 8.6 |
| **Post-admission factors** | | | | | | | | |
| **Number of consultant episodes in spell** | | | | | | | | |
| 1 | 70,029 | 44.5 | 463,175 | 54.7 | 104,635 | 46.5 | 496,509 | 54.8 |
| 2 | 55,680 | 35.4 | 248,449 | 29.4 | 78,466 | 34.8 | 267,575 | 29.5 |
| 3 | 20,691 | 13.2 | 85,012 | 10 | 28,501 | 12.7 | 92,634 | 10.2 |
| 4 or more | 10,822 | 6.9 | 49,674 | 5.9 | 13,660 | 6.1 | 50,115 | 5.5 |
| **Discharge destination** | | | | | | | | |
| Usual residence | 119,024 | 75.7 | 732,906 | 86.6 | 171,005 | 75.9 | 778,821 | 85.9 |
| Other hospital | 7,363 | 4.7 | 37,582 | 4.4 | 9,760 | 4.3 | 39,990 | 4.4 |
| Residential care/care home | 13,022 | 8.3 | 15,239 | 1.8 | 21,260 | 9.4 | 23,608 | 2.6 |
| Died in hospital | 14,364 | 9.1 | 50,493 | 6 | 17,456 | 7.7 | 49,037 | 5.4 |
| Other/unknown/NA | 3,449 | 2.2 | 10,090 | 1.2 | 5,781 | 2.6 | 15,377 | 1.7 |

**Table 3. Summary of samples and hospital outcomes.**

| | Males | | | | | | Females | | | | | |
|---|---|---|---|---|---|---|---|---|---|---|---|---|
| | 2010/11 | | 2012/13 | | 2016/17 | | 2010/11 | | 2012/13 | | 2016/17 | |
| | PwD | PwoD | PwD | PwoD | PwD | PwoD | PwD | PwoD | PwD | PwoD | PwD | PwoD |
| **Length of stay (days)** | | | | | | | | | | | | |
| Number of spells[a] | 87,924 | 659,615 | 111,846 | 706,090 | 157,222 | 846,310 | 153,131 | 756,483 | 182,956 | 786,434 | 225,262 | 906,833 |
| Mean | 10.6 | 7.0 | 10.0 | 6.7 | 9.7 | 6.2 | 10.6 | 7.8 | 10.0 | 7.3 | 9.6 | 6.7 |
| SD | 15.7 | 11.3 | 14.3 | 10.5 | 14.9 | 10.5 | 15.4 | 12.0 | 13.9 | 11.1 | 14.5 | 10.9 |
| Median | 5 | 3 | 5 | 3 | 4 | 3 | 5 | 4 | 5 | 3 | 4 | 3 |
| IQR | 1, 13 | 1, 8 | 1, 13 | 1, 8 | 1, 12 | 1, 7 | 1, 14 | 1, 10 | 1, 13 | 1, 9 | 1, 12 | 1, 8 |
| Maximum | 211 | 314 | 231 | 265 | 304 | 292 | 264 | 336 | 227 | 348 | 272 | 282 |
| **Length of stay> = 15 days** | | | | | | | | | | | | |
| Number of spells[a] | 87,924 | 659,615 | 111,846 | 706,090 | 157,222 | 846,310 | 153,131 | 756,483 | 182,956 | 786,434 | 225,262 | 906,833 |
| Number with the outcome | 20,215 | 88,022 | 24,537 | 88,614 | 31,747 | 96,935 | 36,309 | 119,201 | 41,223 | 115,340 | 46,691 | 118,250 |
| % | 23.0 | 13.3 | 21.9 | 12.5 | 20.2 | 11.5 | 23.7 | 15.8 | 22.5 | 14.7 | 20.7 | 13.0 |
| **Emergency re-admission within 30 days** | | | | | | | | | | | | |
| Number of spells[a][b] | 71,885 | 558,879 | 92,756 | 603,625 | 132,585 | 729,461 | 127,605 | 643,531 | 154,340 | 674,366 | 192,624 | 785,113 |
| Number with the outcome | 18,493 | 100,191 | 23,246 | 105,542 | 35,272 | 148,432 | 28,142 | 104,322 | 33,764 | 107,822 | 43,948 | 143,682 |
| % | 25.7 | 17.9 | 25.1 | 17.5 | 26.6 | 20.3 | 22.1 | 16.2 | 21.9 | 16.0 | 22.8 | 18.3 |
| **Death (in hospital or within 30 days)** | | | | | | | | | | | | |
| Number of spells[a][b] | 85,944 | 644,771 | 111,131 | 701,059 | 156,713 | 841,535 | 149,619 | 733,083 | 181,882 | 779,772 | 224,575 | 900,149 |
| Number with the outcome[c] | 14,010 (10,185) | 62,314 (48,733) | 17,267 (11,958) | 64,662 (49,257) | 21,382 (14,364) | 66,526 (50,493) | 21,788 (14,928) | 65,550 (52,192) | 24,731 (16,200) | 66,457 (51,445) | 27,774 (17,456) | 63,933 (49,037) |
| % | 16.3 | 9.7 | 15.5 | 9.2 | 13.6 | 7.9 | 14.6 | 8.9 | 13.6 | 8.5 | 12.4 | 7.1 |
| **Emergency re-admission or death** | | | | | | | | | | | | |
| Number of spells[a][b] | 80,941 | 597,325 | 104,765 | 650,638 | 146,996 | 776,835 | 140,456 | 677,786 | 170,700 | 722,053 | 210,144 | 829,239 |
| Number with the outcome | 32,471 | 162,414 | 40,501 | 170,151 | 56,635 | 214,833 | 49,905 | 169,793 | 58,477 | 174,215 | 71,702 | 207,551 |
| % | 40.1 | 27.2 | 38.7 | 26.2 | 38.5 | 27.7 | 35.5 | 25.1 | 34.3 | 24.1 | 34.1 | 25.0 |

[a]Excludes spells without a discharge date

[b]Number of spells varied by outcome depending upon extent of follow-up post-discharge and availability of a mortality flag

[c]Number in brackets is deaths in hospital

relative risks (RRs) for males and females by financial year–are given for selected models: (a) with no control for covariates (model M1); (b) with control for all pre-admission factors (M5); (c) with further control for the post-admission factors (M6 or M7). Table A3 in S1 File provides results for all seven steps in the hierarchical analysis. Fig 2 and Fig A1B and A1C in S1 File graph the results for all seven steps, expressed in the form of the "excess" risk for patients with dementia compared to those without.

## Length of stay

In unadjusted analysis, male PwD experienced spells 50% to 55% longer on average than male PwoD in all FYs. Mean spells for female PwD were 36% to 37% longer in 2010 and 2012, but

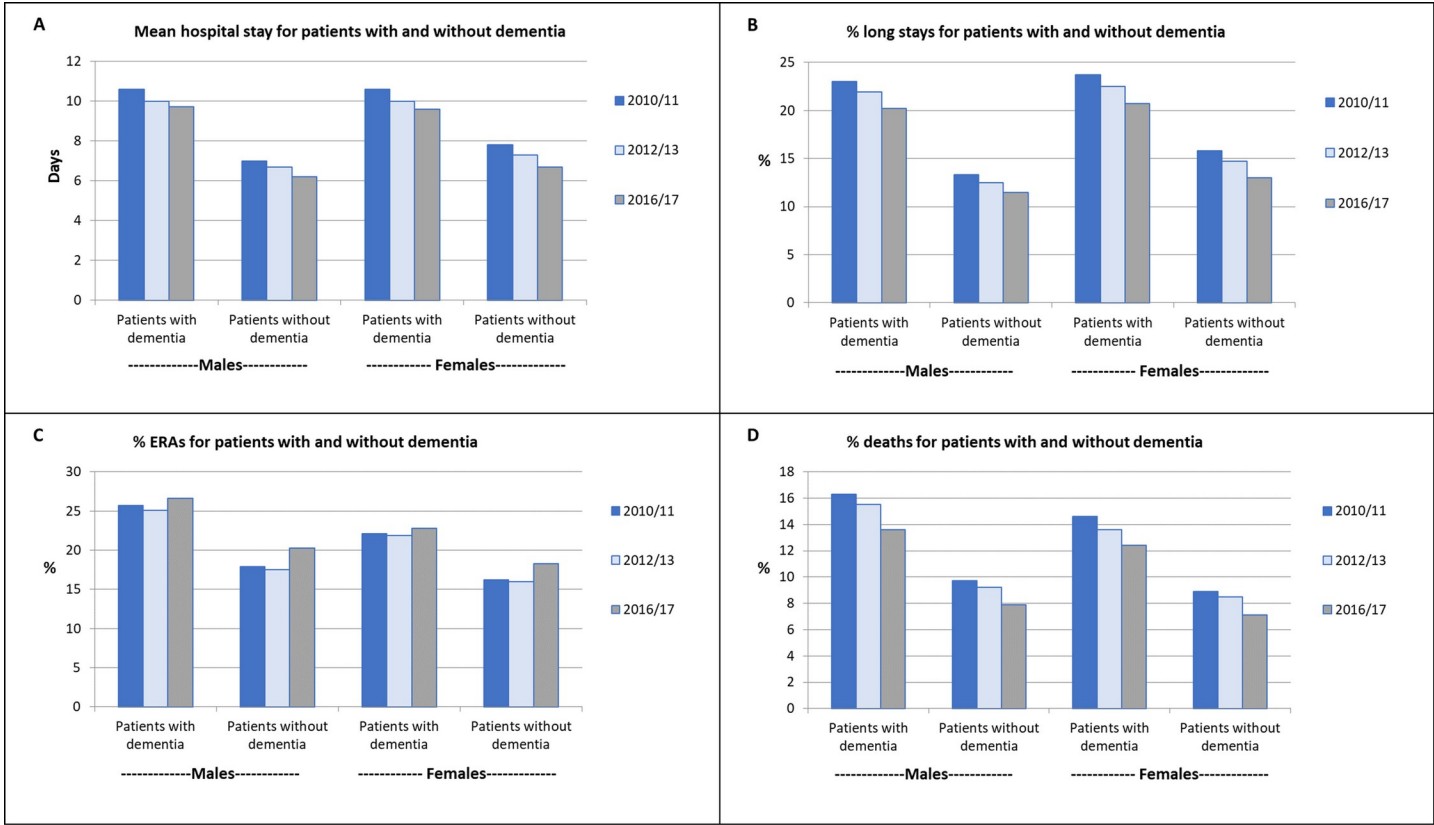

**Fig 1. Unadjusted hospital outcomes for patients with and without dementia.**

43% longer in 2016 (Table 4; Fig 2A and 2B). However, these differences diminished with increasing adjustment for covariates. After inclusion of all pre-admission covariates, the excess mean length of spell was 29% to 35% for males and 19% to 26% for females in all FYs. This reduced further with inclusion of the post-admission covariates, to around 15% for males and 10% for females across the period.

Unadjusted rates of long spells ($> = 15$ days) for PwD were 72% to 76% higher for males and 50% to 59% higher for females in all FYs (Table 4; Fig 2C and 2D). Differences reduced substantially upon adjustment for patient demographics, then more slowly with the addition of further covariates. With all pre-admission covariates included, males were 33% to 38% more likely to have a long stay, and females 20% to 26% more likely. Inclusion of post-admission covariates resulted in further reductions to around 20% for males and 15% for females throughout.

**Emergency re-admissions.** Unadjusted rates of ERAs within 30 days were 43% to 44% higher for male PwD and 36% to 37% higher for female PwD in 2010/11 and 2012/13 respectively, but in 2016/17 just 31% and 25% higher respectively. In all three FYs the group difference in ERA rates decreased as more pre-admission covariates were added, but changed only a little with inclusion of post-admission factors, while remaining lower for 2016/17 throughout (Fig 2E and 2F). In the fully adjusted models for 2016/17 the ERA rate was just 17% higher for both males and females with dementia. Although this represents a reduced group difference compared to earlier FYs, this was driven by an increase in ERAs for PwoD, rather than a reduction for PwD (Table 3).

**Table 4. Summary of group comparisons from hierarchical regression analysis.**

| | Males | | | Females | | |
|---|---|---|---|---|---|---|
| | **2010/11** | **2012/13** | **2016/17** | **2010/11** | **2012/13** | **2016/17** |
| | **RR[a] (95% CI)** | **RR (95% CI)** | **RR (95% CI)** | **RR (95% CI)** | **RR (95% CI)** | **RR (95% CI)** |
| **Length of stay (count in days)** | | | | | | |
| Uncontrolled (M1) | 1.52 (1.49–1.55) | 1.50 (1.47–1.53) | 1.55 (1.52–1.58) | 1.37 (1.34–1.40) | 1.36 (1.33–1.39) | 1.43 (1.40–1.46) |
| Adjusted for pre-admission factors (M5) | 1.31 (1.29–1.33) | 1.29 (1.27–1.32) | 1.35 (1.33–1.38) | 1.20 (1.18–1.22) | 1.19 (1.17–1.21) | 1.26 (1.23–1.28) |
| Adjusted for pre- and post-admission factors (M7) | 1.15 (1.12–1.17) | 1.13 (1.12–1.15) | 1.17 (1.15–1.18) | 1.09 (1.07–1.11) | 1.09 (1.07–1.10) | 1.12 (1.10–1.14) |
| **Length of stay> = 15 days** | | | | | | |
| Uncontrolled (M1) | 1.72 (1.68–1.76) | 1.75 (1.70–1.79) | 1.76 (1.72–1.80) | 1.50 (1.47–1.54) | 1.54 (1.50–1.57) | 1.59 (1.55–1.63) |
| Adjusted for pre-admission factors (M5) | 1.33 (1.30–1.35) | 1.35 (1.32–1.38) | 1.38 (1.36–1.41) | 1.20 (1.18–1.22) | 1.22 (1.20–1.25) | 1.26 (1.23–1.28) |
| Adjusted for pre- and post-admission factors (M7) | 1.19 (1.17–1.22) | 1.20 (1.18–1.23) | 1.22 (1.19–1.24) | 1.13 (1.11–1.15) | 1.15 (1.13–1.17) | 1.16 (1.14–1.18) |
| **Emergency re-admission within 30 days** | | | | | | |
| Uncontrolled (M1) | 1.44 (1.40–1.47) | 1.43 (1.41–1.46) | 1.31 (1.28–1.33) | 1.36 (1.34–1.39) | 1.37 (1.34–1.40) | 1.25 (1.23–1.27) |
| Adjusted for pre-admission factors (M5) | 1.27 (1.25–1.30) | 1.24 (1.22–1.26) | 1.16 (1.14–1.17) | 1.26 (1.24–1.28) | 1.24 (1.22–1.26) | 1.16 (1.14–1.17) |
| Adjusted for pre- and post-admission factors (M7) | 1.29 (1.26–1.31) | 1.25 (1.23–1.27) | 1.17 (1.16–1.19) | 1.28 (1.26–1.30) | 1.25 (1.23–1.27) | 1.17 (1.16–1.19) |
| **Death (in hospital or within 30 days discharge)** | | | | | | |
| Uncontrolled (M1) | 1.69 (1.65–1.72) | 1.68 (1.65–1.72) | 1.73 (1.69–1.76) | 1.63 (1.60–1.66) | 1.60 (1.56–1.63) | 1.74 (1.71–1.77) |
| Adjusted for pre-admission factors (M5) | 1.34 (1.32–1.37) | 1.34 (1.31–1.37) | 1.33 (1.31–1.36) | 1.33 (1.30–1.36) | 1.30 (1.27–1.32) | 1.37 (1.34–1.39) |
| Adjusted for pre- and post-admission factors (M6) | 1.33 (1.31–1.36) | 1.32 (1.30–1.35) | 1.30 (1.28–1.33) | 1.32 (1.30–1.35) | 1.29 (1.27–1.32) | 1.35 (1.33–1.38) |
| **Death in hospital** | | | | | | |
| Uncontrolled (M1) | 1.57 (1.53–1.61) | 1.53 (1.50–1.57) | 1.53 (1.50–1.57) | 1.41 (1.38–1.45) | 1.35 (1.32–1.39) | 1.43 (1.40–1.46) |
| Adjusted for pre-admission factors (M5) | 1.23 (1.20–1.26) | 1.19 (1.16–1.22) | 1.15 (1.13–1.18) | 1.16 (1.13–1.19) | 1.09 (1.07–1.12) | 1.12 (1.10–1.15) |
| Adjusted for pre- and post-admission factors (M6) | 1.22 (1.19–1.25) | 1.18 (1.15–1.21) | 1.14 (1.11–1.16) | 1.16 (1.13–1.19) | 1.09 (1.07–1.12) | 1.11 (1.09–1.14) |
| **Death within 30 days after discharge** | | | | | | |
| Uncontrolled (M1) | 2.26 (2.17–2.35) | 2.33 (2.23–2.43) | 2.48 (2.38–2.58) | 2.64 (2.53–2.75) | 2.53 (2.42–2.64) | 2.86 (2.77–2.96) |
| Adjusted for pre-admission factors (M5) | 1.97 (1.88–2.06) | 1.97 (1.89–2.05) | 1.99 (1.91–2.08) | 2.20 (2.12–2.30) | 2.08 (1.98–2.17) | 2.27 (2.19–2.35) |
| Adjusted for pre- and post-admission factors (M6) | 1.94 (1.85–2.03) | 1.93 (1.85–2.01) | 1.94 (1.86–2.02) | 2.20 (2.11–2.29) | 2.06 (1.97–2.16) | 2.23 (2.15–2.31) |
| **Emergency re-admission or death** | | | | | | |
| Uncontrolled (M1) | 1.48 (1.46–1.50) | 1.48 (1.46–1.50) | 1.39 (1.38–1.41) | 1.42 (1.40–1.43) | 1.42 (1.40–1.44) | 1.36 (1.35–1.38) |
| Adjusted for pre-admission factors (M5) | 1.28 (1.26–1.30) | 1.26 (1.25–1.28) | 1.21 (1.20–1.22) | 1.27 (1.25–1.28) | 1.25 (1.24–1.27) | 1.22 (1.21–1.23) |
| Adjusted for pre- and post-admission factors (M6) | 1.28 (1.26–1.29) | 1.26 (1.24–1.27) | 1.20 (1.19–1.22) | 1.27 (1.25–1.28) | 1.25 (1.24–1.26) | 1.22 (1.21–1.23) |

[a]RR = Relative Risk (strictly, the Incident Rate Ratio)

**Deaths.** Unadjusted rates of deaths were 68% to 73% higher for male PwD in all three FYs. This reduced to 33% to 36% after adjustment for patient demographics but changed only slightly as additional covariates were added (Fig 2G). The pattern for female PwD was similar except that excess deaths were somewhat higher in 2016/17, at a fully adjusted excess rate of 35% (Table 4; Fig 2H). Most deaths occurred in hospital: in 2016/17 8.3% of PwD compared to 5.7% of PwoD died in hospital, with smaller percentages, 4.5% and 1.8%, dying after discharge. However, after full covariate adjustment, excess rates of deaths in hospital were just 14% and 11% for male and female PwD respectively. By contrast, amongst those discharged, excess rates of deaths for PwD were much higher, at 94% and 123% respectively. A small reduction in excess deaths in hospital since 2010/11 was somewhat counter-balanced by an increase in excess deaths post-discharge (Table 4; Fig 3).

**Emergency readmission or death.** When analysed as a combined outcome, excess rates of ERAs plus deaths for PwD were a little lower in 2016/17 compared to previous FYs for both sexes. Rates showed a sharp decline for both sexes in all three FYs after adjustment for patient

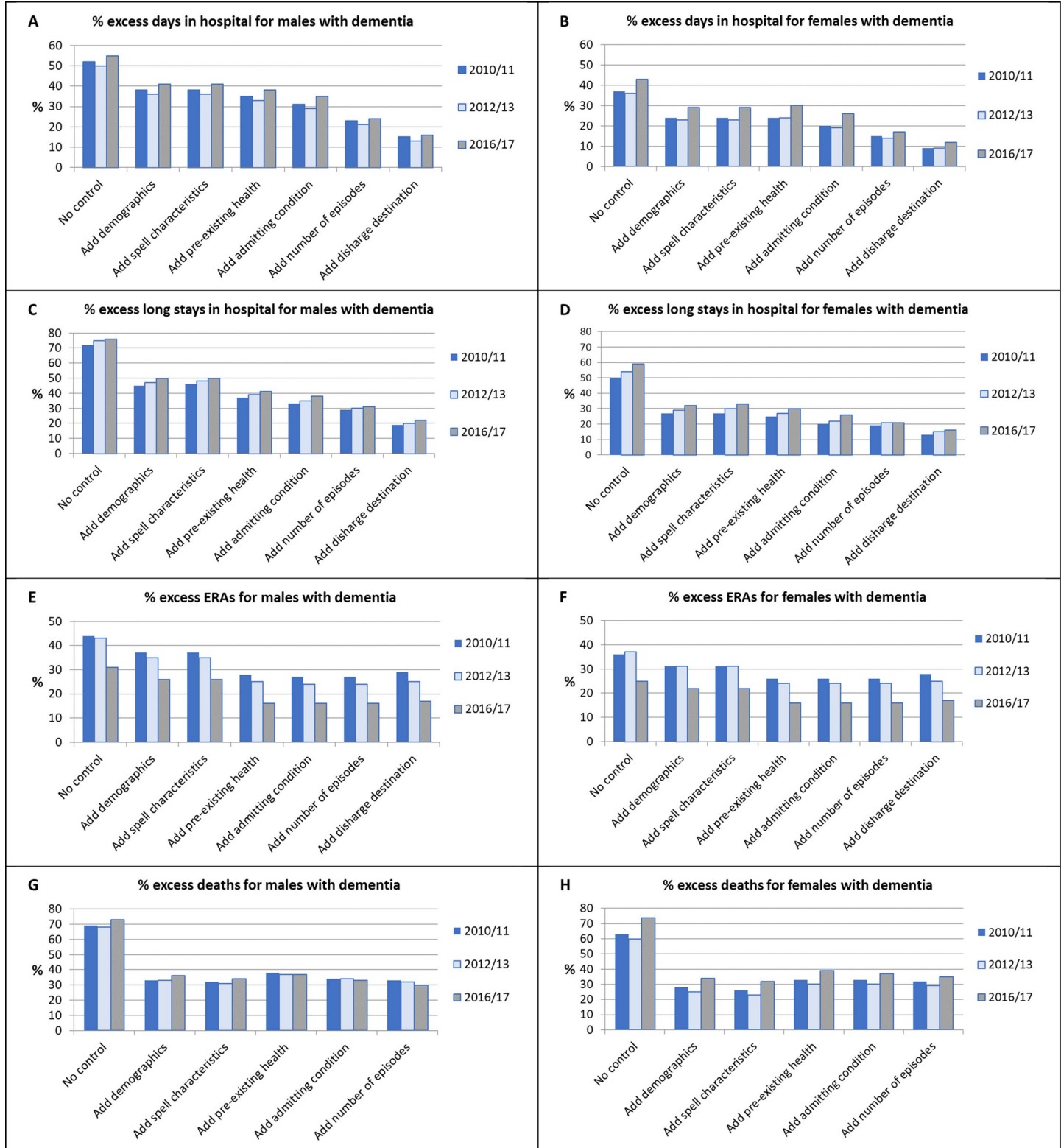

**Fig 2. Patient group differences resulting from each step in the hierarchical regression analysis.** For death as an outcome, discharge destination was excluded from the models.

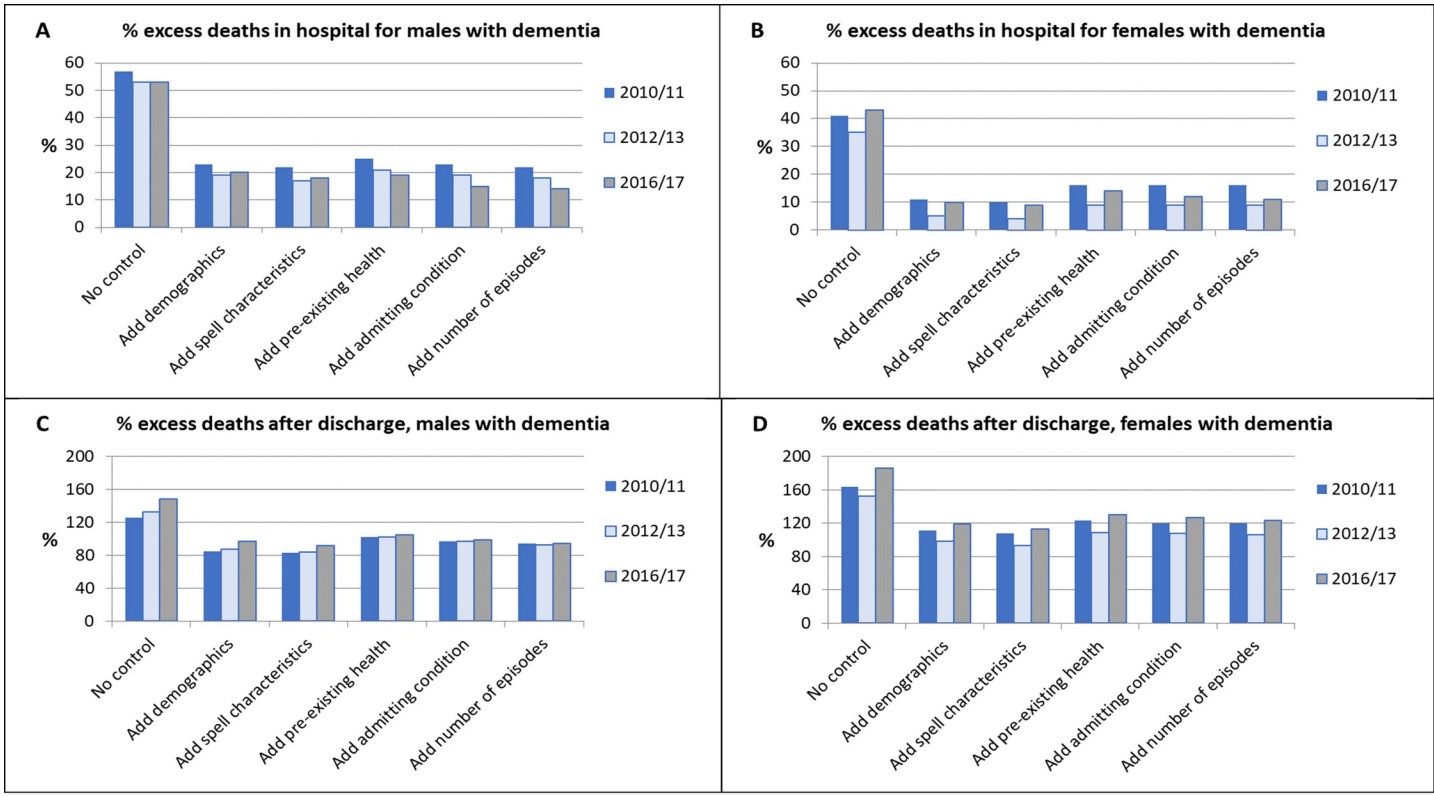

**Fig 3. Patient group differences resulting from each step in the hierarchical regression analysis for deaths in hospital and deaths post-discharge.** For death as an outcome, discharge destination was excluded from the models.

demographics, followed by a more gradual decline with increasing covariate control (Table A3 in S1 File; Fig A1B and A1C in S1 File).

**Sensitivity analysis.** Findings of the sensitivity analyses were very similar across FYs so only 2016 results are reported here. Excluding hospital spells of <2 days, the fully adjusted excess length of stay for PwD increased slightly to 23% for males and 17% for females; adjusted results for other outcomes hardly changed (Table A4 in S1 File; Fig A1D and Fig A2 in S1 File). Exclusion of spells involving a psychiatric comorbidity increased the group difference for all outcomes, though by only a few percentage points. Results using the matched patient sample closely corresponded to the primary analysis with the exception of excess ERAs, which increased to 25% for males and 21% for females under full covariate adjustment. Use of a logistic model to analysis the binary outcomes made no appreciable difference. Use of a random-effects model produced substantially lower estimates of excess LoS, to just 9% for male PwD and 6% for female PwD under full adjustment; results from random-effects models for other outcomes closely matched the primary analysis. Sub-hazard ratios from the competing risks survival analysis estimated that after adjusting for all covariates and for the risk of dying in hospital, at any point in a spell males with dementia were 11% less likely to be discharged and females 8% less.

## Discussion

In unadjusted analysis PwD of both sexes experienced longer LoS and higher rates of ERAs and mortality across the six year period. However, most of the apparent disparity in outcomes

could be accounted for by group differences in patient and spell characteristics. After full adjustment, LoS in 2016/17 was on average just 17% longer for male PwD and 12% longer for female PwD, equivalent to an average of 1.0 and 0.8 additional days in hospital, and nearly half of that under random-effects and competing risks survival models. These adjusted rates remained virtually constant across the six years. Results were very similar for patients experiencing long hospital stays.

The adjusted excess risk of an ERA in 2016/17 was 17% for PwD of both sexes, with post-admission factors making little difference. These rates were lower than in previous FYs, but interpretation is not straight-forward since absolute ERA rates increased in 2016/17 for both groups, except less-so for PwD.

Whilst the large majority of deaths in both patient groups occurred in hospital, after adjustment PwD had only a slightly elevated risk of dying in hospital, whereas their risk of dying within 30 days post-discharge was higher by around 90% for males and 110% for females throughout the study period.

The pattern of results is complex but in essence we found that covariate-adjusted differences between Pwd and PwoD in LoS, ERAs and in-hospital deaths were small throughout the period 2010 to 2016. These findings were remarkably consistent across the sexes, adding to their validity. Furthermore, although we adjusted for multiple covariates, many other factors remain that we could not quantify or control that may account for much, if not all, of the remaining group differences. PwD are often more vulnerable upon admission, for example presenting with malnourishment [7], delirium [36, 37] and frailty; conditions only partially captured in our analysis that increase the risk of poorer outcomes [9, 38–40]. They are also less able to adapt to the hospital environment, resulting in significant distress and deterioration [41, 42]. Such factors engender group differences that hospitals have limited ability to affect.

An important exception was our finding that PwD were around twice as likely to die shortly after discharge, despite having a similar rate of death whilst in hospital. This might suggest that PwD were more likely to be discharged before fully ready, or that they had vulnerabilities that were being managed in the hospital environment, but which when compounded with dementia impacted negatively on recovery post-discharge. However, under these conditions one would also expect to see a high excess rate of ERAs, which is not present. Difficulties in identifying the terminal phase of dementia make PwD much more likely to be admitted than offered alternatives such as referral to palliative care services [42, 43], hence this finding may possibly reflect individuals in the terminal stages of dementia being discharged to die at home upon their terminal condition becoming apparent.

The many national and local care initiatives during this period saw hospitals, charities and other organisations invest considerable effort and resources into improving inpatient care for PwD and in expanding staff training. Yet it seems that the widespread assumption that this activity would impact on perceived disparities in the outcomes investigated in this study was mis-placed, since we find those disparities to be mostly–if not entirely—a matter of patient case-mix differences. Quite possibly the factors that most influence these particular outcomes are more a matter of service capacity than of hospital culture, such as staff shortages, pressure to free up beds and the availability of community placements, which can affect all patients not only PwD [44–46]. We note that the many initiatives may well have resulted in other kinds of benefits: NAD surveys across the period using organisational-level checklists and staff and patient/carer questionnaires, show that hospital scores against a set of standards relating to patient assessments, information and communication, staffing and training, nutrition, discharge and transfer, and governance, all improved substantially [47]. In addition the policy and other initiatives were very much focused on improving the inpatient experience for PwD, such as improving communications and interactions with staff and reducing potential distress

and discomfort. Patient/carer experience is an important aspect of care not captured within the APC and hence not available to our study.

## Implications for policy and practice

Less emphasis should be given to seeking parity in LoS and ERAs for PwD, for which covariate adjusted disparities were small and may even represent uncontrolled confounding. The same is true for mortality in hospital, which accounts for the large majority of all deaths. By contrast the much higher rates of post-discharge mortality amongst PwD require more attention [46]. This may reflect individuals in the terminal stages of dementia being discharged to die at home once their terminal condition becomes apparent, suggesting a need for better identification of such cases before admission with a view to providing suitable alternatives to hospital care [48, 49]. Ensuring that hospital staff training includes recognition of the terminal phase of dementia and end-of-life care could help facilitate this. Another contributing factor might be a negative impact of dementia on recovery post-discharge, for example due to reduced ability to self-care, implying a need for greater post-discharge support from community services.

## Implications for research

Research is needed to better understand the much higher rate of post-discharge mortality amongst PwD, especially the role of pre-existing health issues and the impact of dementia on recovery following discharge. One possible approach using hospital data would be an analysis of outcomes following hospital discharge for people with and without dementia, matched or controlled for clinical factors prior to discharge. Outcomes might include length of time until a new admission, mortality, and transfer to residential care. Ideally, data about each individual's hospital discharge plan and the community and social services received would be linked in. Further thought would need to be given as to whether the unit of analysis should be at the discharge or the patient level.

Our findings also question the widespread assumption that group differences in LoS, ERA and mortality are indicative of disparities in the quality of hospital care received. Investigations are needed to clarify the relationships here, and to potentially develop more sensitive outcome measures. Future research needs to take greater account of the many ways in which PwD differ from other inpatients and be aware that even in highly-controlled studies considerable uncertainty is likely to remain over the extent to which group differences reflect genuine variation as opposed to uncontrolled confounding.

## Comparison to previous literature

Möllers et al in a systematic review found that 52 out of 60 observational studies reported a longer hospitalisation time for people with dementia, with mean stays longer by up to 22 days. However, very few studies controlled for confounding factors [10]. An integrative review also reported LoS to be typically longer for PwD, though not in every study, while in the same review 8 out of 11 studies found PwD to be at higher risk of in-hospital mortality, though this varied by various factors including age, presence of delirium, and disease condition [7]. Another more recent systematic review drew attention to wide methodological heterogeneity between studies but concluded that the majority report poorer outcomes for PwD, though the authors refrained from quantifying the differences [37]. A systematic review of cohort and case-control studies of ERA rates concluded that any increased rate due to dementia per se was modest at best, in the range 3% to 13% [50], close to our present estimate of 17%.

Few UK-based studies have controlled for clinical factors. The UK Care Quality Commission (CQC) analysed national APC data on EAs from 2008 to 2012, using patient samples

matched on age, gender, primary diagnosis and co-morbidities [25]. PwD had higher rates for all outcomes across the period though the gap decreased over time: excess LoS declined from 32% to 26%; ERAs from 21% to 18% and risk of death from 49% to 36%. Rates in the present study adjusted for pre-admission factors are comparable but suggest a slowing, or even reversal, in trend.

Two relevant single-site studies report highly variable effect sizes, but being based on incident admissions rather than individual spells cannot be directly compared to our own (see below). Fogg et al analysed 21,399 incident EAs between 2014 and 2017 at one large acute English hospital, with patients screened for cognitive impairment upon admission [23]. Several covariates were controlled. Survival analysis treating in-hospital death as a competing risk estimated that at any given time PwD were 20% (95%CI 17% to 24%) less likely to be discharged. PwD also had an increased risk of readmission within 30 days (OR = 1.21; 95% CI 1.04 to 1.40) and of death in hospital or within 30 days of discharge (OR = 1.66; 95%CI 1.48 to 1.86). An investigation of incident EAs (n = 6724), at one large Scottish district general hospital in 2012 and 2013 [37, 51], compared four "cognitive spectrum disorder" (CSD) groups (dementia, delirium, both, unspecified) to non-CSD patients. Adjusted for baseline demographic and comorbidity covariates, LoS for PwD was 73% (95%CI 54% to 94%) longer than for those without CSD and risk of re-admission was 33% (16% to 52%) higher, but the risk of death within three months of admission was similar (HR = 1.04; 95%CI 0.84 to 1.29).

## Strengths and weaknesses

Our patient samples were many times larger than any previous study and we included all English acute hospitals of any substantial size for three different financial years. We also took account of a larger number and range of confounding factors. Our findings were mostly highly stable across FYs and a variety of sensitivity analyses. They were also very similar for males and females analysed separately. However, our data series finished in March 2017 and analysis of more recent data is required, especially covering the period of the coronavirus pandemic. Our results should be fully representative of the NHS in England across the period but the extent of generalisability to other countries and health systems is uncertain.

Our study is not a patient-level longitudinal analysis in the sense of following a specific cohort of patients with and without dementia across time. Rather, we analysed cross-sectional samples of individual hospital spells (also called prevalent admissions) at three points in time. This facilitated the estimation of group differences at each time-point adjusted for spell-specific clinical factors and comparison across time-points. Other UK-based studies have adopted an incident admission approach, where individual patients are followed over time by combining across sequential hospital spells for the same underlying condition until death, final discharge, or a fixed length of follow-up [23, 51]. Since most spells last only a few days while the periods in-between are often weeks or months, outcomes for incident admissions tend to be dominated by events and care during the periods outside of hospital between spells, including provision from other types of service providers. Although informative about the totality of care received, this approach is less informative about the outcomes of hospital care itself.

We focused on patients with dementia as this was the central policy issue over the period, though similar concerns pertain to patients with other cognitive impairments, most notably delirium, for whom patterns of outcomes are broadly similar [37, 51]. Delirium is greatly under-recorded in the APC, with one UK study finding that routine coding detected only around 10% of the delirium cases diagnosed via a gold standard clinical assessment [52].

Our findings are dependent upon the reliability and completeness of the hospital record, which can be variable [53], and whilst we adjusted for many covariates we were limited by

what was available in the hospital record. Non-mandatory comorbidities are only recorded if deemed clinically relevant and some HRG subchapters are very broad, implying a potential for uncaptured confounding that might further account for group differences. To help adjust for patient complexity and health needs our models included the post-admission factors of number of consultant episodes and discharge destination. These accounted for around half of the group difference in LoS, whilst leaving rates of ERAs and mortality unaffected. For some patients these factors may reflect a potentially avoidable event in hospital, such as a fall or infection leading to an additional consultant episode. Any over-adjustment of LoS resulting from this, will have been balanced to some degree by opposite bias due to uncontrolled confounding factors.

Dementia status was defined by the presence of a related code in the hospital record during the previous five years and some patients may have been mis-classified. Greater mis-classification in earlier years is suggested by the increased representation of both male and female PwD in the patient cohorts over time. There was hardly any change in mean age or other demographics for either PwD or PwoD across the period, so this increase cannot be explained by a changing population and is more likely due to improved case-finding driven by the Dementia CQUIN [12]. Nonetheless, compared against a large mental health registry, APC dementia diagnoses over this period have been found reasonably accurate, with sensitivity of 78% and specificity of 92% over the span of a hospital record [54]. In addition, mis-classification predominantly involves PwD being mistakenly assigned to the PwoD cohort rather than the reverse, and as a very small proportion of that cohort any impact on estimated group differences will be small.

Sensitivity analysis excluding spells involving comorbidities likely to be confused with dementia, produced mostly only small increases in the relative risk estimates. However, we lacked information on the stage of dementia. Hospital admission and discharge dates (used to calculate length of stay and ERAs) are expected to be accurate and deaths were based on Office for National Statistics mortality data. We suspect that the coding of discharge destination may under-represent discharges to residential homes.

In view of model complexity and computational overheads we analysed each FY separately and performed no direct tests for trends or differences across FYs. Sample sizes were such that most RRs were estimated with a 95% CI of plus/minus 2% or 3%, hence even small differences would reach statistical significance. We were not able to take into account wider organisational-level factors that might further explain or moderate patient group differences, such as how research active a Trust was [55] or the availability of community placements and services that might impact on outcomes.

## Conclusion

Over the six-year period, covariate-adjusted hospital LoS, ERA rates and in-hospital mortality rates for PwD were only slightly elevated compared to similar patients without dementia and remaining differences potentially reflect uncontrolled confounding. PwD however, were consistently much more likely to die shortly after discharge, the reasons for which require further investigation. Despite being widely used for service evaluation, LoS, ERA and mortality may lack sensitivity to changes in hospital care and support to PwD.

## Supporting information

**S1 Checklist. STROBE statement—checklist of items that should be included in reports of observational studies.**
(DOCX)

**S1 File. Supplementary materials.**
(DOCX)

## Acknowledgments

We would like to thank the DEMTRAIN study Advisory Group, that included people living with dementia and was Chaired by Dr Dawne Garratt, who was the RCN Professional Lead for Older People and Dementia Care, who have given feedback throughout the study. We particularly want to thank and acknowledge Chloe Hood, Programme Manager, and the National Audit of Dementia team at the Royal College of Psychiatry for their support and advice on matters related to the NAD surveys and datasets. Further thanks go to the members of our Expert Panel (Asan Akpan, Neil Pendleton and Emma Vardy) for their advice on clinical aspects of dementia, related ICD10 codes, and the understanding and interpretation of Hospital Episode Statistics data. We would also like to thank Professor Alistair Burns in his role as National Clinical Director for Dementia and Older People's Mental Health at NHS England and NHS Improvement for his help and assistance in the work programme.

## Author Contributions

**Conceptualization:** David Reeves, John Keady, Siobhan Reilly.

**Data curation:** Fiona Holland.

**Formal analysis:** David Reeves, Fiona Holland, Mark Hann.

**Funding acquisition:** David Reeves, John Keady, Siobhan Reilly.

**Methodology:** David Reeves, Fiona Holland, Hazel Morbey, Mark Hann, Faraz Ahmed, Linda Davies, John Keady, Iracema Leroi, Siobhan Reilly.

**Project administration:** David Reeves, Siobhan Reilly.

**Resources:** David Reeves, Fiona Holland, John Keady.

**Software:** Fiona Holland, Linda Davies.

**Supervision:** David Reeves.

**Writing – original draft:** David Reeves.

**Writing – review & editing:** David Reeves, Fiona Holland, Hazel Morbey, Mark Hann, Faraz Ahmed, Linda Davies, John Keady, Iracema Leroi, Siobhan Reilly.

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
