## [Decision Letter · Decision Letter 0]

28 Nov 2022

PONE-D-22-26774Retrospective study of more than 5 million emergency admissions to hospitals in England: epidemiology and outcomes for people with dementiaPLOS ONE

Dear Dr. Reeves,

Thank you for submitting your manuscript to PLOS ONE. After careful consideration, we feel that it has merit but does not fully meet PLOS ONE’s publication criteria as it currently stands. Therefore, we invite you to submit a revised version of the manuscript that addresses the points raised during the review process. Please submit your revised manuscript by Jan 12 2023 11:59PM. If you will need more time than this to complete your revisions, please reply to this message or contact the journal office at plosone@plos.org. Please include the following items when submitting your revised manuscript:A rebuttal letter that responds to each point raised by the academic editor and reviewer(s). You should upload this letter as a separate file labeled 'Response to Reviewers'.A marked-up copy of your manuscript that highlights changes made to the original version. You should upload this as a separate file labeled 'Revised Manuscript with Track Changes'.An unmarked version of your revised paper without tracked changes. You should upload this as a separate file labeled 'Manuscript'.If applicable, we recommend that you deposit your laboratory protocols in protocols.io to enhance the reproducibility of your results. Protocols.io assigns your protocol its own identifier (DOI) so that it can be cited independently in the future. For instructions see: https://journals.plos.org/plosone/s/submission-guidelines#loc-laboratory-protocols. Additionally, PLOS ONE offers an option for publishing peer-reviewed Lab Protocol articles, which describe protocols hosted on protocols.io. Read more information on sharing protocols at https://plos.org/protocols?utm_medium=editorial-email&utm_source=authorletters&utm_campaign=protocols.

We look forward to receiving your revised manuscript.

Kind regards,

Sreeram V. Ramagopalan

Academic Editor

PLOS ONE

Journal Requirements:

3. Please include your ethics statement in the Methods section of your manuscript. In the Methods section of your revised manuscript, please include the full name of the institutional review board or ethics committee that approved the protocol, the approval or permit number that was issued, and the date that approval was granted.

Reviewers' comments:

Reviewer's Responses to Questions

**Comments to the Author**

1. Is the manuscript technically sound, and do the data support the conclusions?

Reviewer #1: Yes

2. Has the statistical analysis been performed appropriately and rigorously? 

Reviewer #1: Yes

3. Have the authors made all data underlying the findings in their manuscript fully available?

Reviewer #1: Yes

4. Is the manuscript presented in an intelligible fashion and written in standard English?

Reviewer #1: Yes

5. Review Comments to the Author

Reviewer #1: The authors present several analyses regarding emergency hospital admissions and outcomes of people with and without dementia, using a large, representative dataset.

Clarify in the abstract that the one of the relevant outcomes is "Hospital mortality or within 30 days post discharge"

It would be interesting to know the outcomes separated by in-hospital mortality vs out of hospital (happening within 30 days of discharge). This could provide light into where most of the excess deaths happened and point to different hypothesis and potential interventions. I wonder if the competing risk analysis had this intent, however I am not clear on one point of that analysis (see next point)

“Sub-hazard ratios from the competing risks survival analysis estimated that after full adjustment males with dementia were 11% less likely to be discharged and females 8% less, at any point in time.” -> can you please expand on this sentence? Is this statement about in hospital death? Previously it was indicated that this competing risk analysis was about death any time post 30 days competing with ERA.

Explanatory factors point to a causal analysis, I suggest it is renamed either “Covariables” or “Associated Factors”. In the same direction, other references to “explanation” in the manuscript and replace for terms that point to the hypothesis generating nature of this research.

I suggest authors indicate that this is not a complete patient level longitudinal analysis in the sense of following a cohort of patients with dementia diagnosis at admission and a group not having dementia. The multi-level structure of the data stems from patients being clustered within hospitals, from short term 30 days follow-up and because previous # of inpatient episodes were identified. But there is no analysis of the same patient being tracked over certain continuous (or discrete but jointed) time which could point to patients with particular difficult and/or actionable trajectories. This does not diminish the merit of the paper, but it would be interesting if the authors comment if such analysis could be useful and whether this could be addressed linking HES with other sources or if some other datasets are needed. Authors could include some thoughts as to whether a multi-year patient level longitudinal analysis of the same or similar outcomes could shed more light in the future. Researchers could also comment if such dataset exists (and point to references), or will exist in the future (and point to project plans if public), or, if none of the previous, stress the importance of funding such efforts.

A limitation that should be commented is that that the last period analyzed is already around 5 years ago. Looking at https://bmjopen.bmj.com/content/bmjopen/8/3/e020325.full.pdf it does seem that more recent years could have been analyzed. Covid period comes to mind as an immediate time where the observed trends may have worsened for PwD.

I agree with the conclusion that findings are of practically no increased risk for many of the outcomes analyzed, or at most very mild measures of increased relative risk that cannot exclude residual confounding. I wonder, given the very small relative risk at the initial period, particularly after accounting for other factors, if it would have been expected to see a reduction because the starting point is already quite similar. Authors comment about looking for potentially more sensitive outcome measures which I also find interesting.

6. PLOS authors have the option to publish the peer review history of their article (what does this mean?). If published, this will include your full peer review and any attached files.

Reviewer #1: No

---

## [Author Response · Author response to Decision Letter 0]

12 Jan 2023

We confirm that our manuscript meets these requirements

The study is based on Hospital Episode Statistics (HES) data routinely collected from all secondary care providers in England, with a link to Office of National Statistics death data. All data is pseudonymised for the purposes of research. Release of the data does not require Ethics Board approval or informed patient consent. Instead, release of HES and Office of National Statistics data is via application to the NHS Digital Data Access Request Service (DARS) subject to approval by the Independent Group Advising on the Release of Data (IGARD). This study gained approval on 6/4/2017, DARS reference NIC 33318.

We have accordingly dropped our Ethics statement and revised the text in the Methods section, as stated below.

3. Please include your ethics statement in the Methods section of your manuscript. In the Methods section of your revised manuscript, please include the full name of the institutional review board or ethics committee that approved the protocol, the approval or permit number that was issued, and the date that approval was granted.

We have dropped the Ethics statement since Ethics approval was not required nor sought.

We have revised the text in the Methods section as below:

“This study is based on the English National Health Service (NHS) Hospital Episode Statistics (HES) database, linked to Office of National Statistics (ONS) official death data and pseudonymised for the purposes of research. The data was obtained via application to the NHS Digital Data Access Request Service (DARS) on 6/4/2017, DARS reference NIC 33318. Specifically, We extracted obtained extracts of individual patient data from the admitted patient care (APC) dataset within HES for the financial years (FYs; UK government accounting years, which run from April 1st to the following March 31st) 2010/11, 2012/13 and 2016/17.”

The conditions under which we obtained the data from NHS Digital do not allow us to share it publically. We now include the below Data Availability statement. The wording has been adapted from Data Availability statements that appear in existing papers published in PLOS ONE that make use of the HES database.

Data Availability: Electronic health records are, by definition, considered sensitive data in the UK by the Data Protection Act and cannot be shared via public deposition because of information governance restrictions in place to protect patient confidentiality. For researchers who meet the criteria for access to confidential data, the Hospital Episode Statistics and linked Office of National Statistics mortality data used for this study is available via application to the UK Health and Social Care Information Centre. For details see: https://digital.nhs.uk/data-and-information/data-tools-and-services/data-services/hospital-episode-statistics

Please see our response above

We have dropped the Ethics statement since Ethics approval was not required nor sought. However, we have added more detail to the Methods section about the source of the data we analysed: 

“This study is based on the English National Health Service (NHS) Hospital Episode Statistics (HES) database, linked to Office of National Statistics (ONS) official death data and pseudonymised for the purposes of research. The data was obtained via application to the NHS Digital Data Access Request Service (DARS) on 6/4/2017, DARS reference NIC 33318. Specifically, We extracted obtained extracts of individual patient data from the admitted patient care (APC) dataset within HES for the financial years (FYs; UK government accounting years, which run from April 1st to the following March 31st) 2010/11, 2012/13 and 2016/17.”

We have checked our reference list and confirm that it is complete and correct

5. Review Comments to the Author

Reviewer #1: The authors present several analyses regarding emergency hospital admissions and outcomes of people with and without dementia, using a large, representative dataset.

Clarify in the abstract that the one of the relevant outcomes is "Hospital mortality or within 30 days post discharge"

Done

It would be interesting to know the outcomes separated by in-hospital mortality vs out of hospital (happening within 30 days of discharge). This could provide light into where most of the excess deaths happened and point to different hypothesis and potential interventions. 

This is an excellent suggestion that we thank the reviewer for and have now taken up. The results show that rates of in-hospital deaths differed only a little between patient groups but that the post-discharge mortality rate for PwD was around twice the rate for controls. This has had implications for the manuscript text and interpretation of results in several places. The results of this new analysis are summarised in the results section (from line 312) and the results are tabulated in Table 4 and Table A3 in the supplementary file and presented in the main manuscript in a new Figure 3.The previous Figure 3, summarising results for the death+ERA outcome, now appears in the supplementary file. Discussion of the new findings is presented at line 371 and the sections on implications for policy and for research, along with our conclusion, have been revised accordingly.

I wonder if the competing risk analysis had this intent, however I am not clear on one point of that analysis (see next point)

“Sub-hazard ratios from the competing risks survival analysis estimated that after full adjustment males with dementia were 11% less likely to be discharged and females 8% less, at any point in time.” -> can you please expand on this sentence? Is this statement about in hospital death? Previously it was indicated that this competing risk analysis was about death any time post 30 days competing with ERA.

The statement refers to the competing risk analysis for length of stay where death in hospital was the competing risk. This is described at line 205 of the manuscript. We realise that we have caused some confusion by also referring to ERA and Death as “competing outcomes”. Use of the term “competing” here was meant to indicate that a hospital spell cannot result in both an ERA and Death: if changes in care, say, result in fewer deaths this might lead to an increase in the ERA rate (more patients live to experience an ERA). To assess this we analysed the combined outcome of ERA OR Death. We have revised the wording about this combined outcome in the Outcomes section to remove the term “competing” to avoid confusion.

We have also rephrased the sentence at line 340 to be clearer on interpretation of the sub-hazard ratios: “Sub-hazard ratios from the competing risks survival analysis estimated that after adjusting for all covariates and for the risk of dying, at any time during a spell males with dementia were on average 11% less likely to be discharged and females 8% less.”

Explanatory factors point to a causal analysis, I suggest it is renamed either “Covariables” or “Associated Factors”. In the same direction, other references to “explanation” in the manuscript and replace for terms that point to the hypothesis generating nature of this research.

We have replaced the term “explanatory factors” by “clinical factors”, using this in a broad sense to include patient demographics and health as well as medical procedures. We have also amended the text to avoid suggestions of a causal relationship. 

I suggest authors indicate that this is not a complete patient level longitudinal analysis in the sense of following a cohort of patients with dementia diagnosis at admission and a group not having dementia. The multi-level structure of the data stems from patients being clustered within hospitals, from short term 30 days follow-up and because previous # of inpatient episodes were identified. But there is no analysis of the same patient being tracked over certain continuous (or discrete but jointed) time which could point to patients with particular difficult and/or actionable trajectories. This does not diminish the merit of the paper, but it would be interesting if the authors comment if such analysis could be useful and whether this could be addressed linking HES with other sources or if some other datasets are needed. Authors could include some thoughts as to whether a multi-year patient level longitudinal analysis of the same or similar outcomes could shed more light in the future. Researchers could also comment if such dataset exists (and point to references), or will exist in the future (and point to project plans if public), or, if none of the previous, stress the importance of funding such efforts.

We have added text to the Discussion (line 467) explaining the differences between our spell-level analysis and a patient-level longitudinal model. This has helped us to make it clearer that our approach is focused – as far as possible given the nature of the data – on outcomes in relation to the care provided in hospital. This is substantially different to the type of patient-level analysis proposed by the reviewer, known as an “incident admissions” approach in this field, which is more focused on the totality of provided care. We have added the text:

“Our study is not a patient-level longitudinal analysis in the sense of following a specific cohort of patients with and without dementia across time. Rather, we analysed cross-sectional samples of individual hospital spells (also called prevalent admissions) at three points in time. This facilitated the estimation of group differences at each time-point adjusted for spell-specific clinical factors and comparison across time-points. Other UK-based studies have adopted an incident admission approach, where individual patients are followed over time by combining across sequential hospital spells for the same underlying condition until death, final discharge, or a fixed length of follow-up.(23, 56) Since most spells last only a few days while the periods in-between are often weeks or months, outcomes for incident admissions tend to be dominated by events and care during the periods outside of hospital between spells, including provision from other types of service providers. Although informative about the totality of care received, this approach is less informative about the outcomes of hospital care itself.”

With regard to the reviewer’s query as to whether such an analysis would be useful, that would be true if the objective was to evaluate outcomes in relation to the totality of received care both in and out of hospital. However, that was not the objective of our study. In addition, while group differences in outcomes may be found, there would be no way of partialing out the relative contributions of the hospital and out-of-hospital components, or of the different providers that may have been involved. Studies we know of that have used the incident admissions approach tend to implicitly ascribe resulting group differences to deficiencies in the initial hospital care, forgetting about the impact of the care received during the periods between spells. 

We also have some statistical reservations about the implementation of the incident admission approach which we have not gone into in our manuscript due to space. Incident admissions are not easy to identify in HES so researchers typically set a “wash-out” period. For example, Reynish et al defined an admission as incident if no prior admission had occurred in the previous 6 months, resulting in the exclusion of nearly half of all admitted patients. This will have biased their sample towards patients with less frequent admissions, who also tend to be less complex. Also, covariate adjustment in such studies has been restricted to clinical factors for the first spell only: although incorporating time-varying co-variates may be possible no study has attempted this and it would add another level of complexity to an already quite complex analytical model.

The reviewer’s comment has encouraged us to think more deeply about the issues here and the implications for research. We have accordingly revised the passage on future research (line 412). The large group difference in post-discharge mortality along with little difference in any other outcome, points to the post-discharge period as key. We now suggest that hospital data could be used to investigate this through an analysis of discharges (rather than admissions). As the reviewer suggests, such an analysis would benefit greatly from linkage to other data, such as details of post-discharge plans and the involvement of community care and social services. Whether such an analysis would be more appropriately conducted at the discharge or the patient level requires considerable further thought and we would not like to commit at this stage, so have left this open. Likewise, we cannot at this stage suggest a specific dataset, or datasets capable of linkage, that would provide all the key relevant information:

“Research is needed to better understand the much higher rate of post-discharge mortality amongst PwD, especially the role of pre-existing health issues and the impact of dementia on recovery following discharge. One possible approach using hospital data would be an analysis of outcomes following hospital discharge for people with and without dementia, matched or controlled for clinical factors prior to discharge. Outcomes might include length of time until a new admission, mortality, and transfer to residential care. Ideally, data about each individual’s hospital discharge plan and the community and social services received would be linked in. Further thought would need to be given as to whether the unit of analysis should be at the discharge or the patient level.”

A limitation that should be commented is that that the last period analyzed is already around 5 years ago. Looking at https://bmjopen.bmj.com/content/bmjopen/8/3/e020325.full.pdf it does seem that more recent years could have been analyzed. Covid period comes to mind as an immediate time where the observed trends may have worsened for PwD.

We entirely agree with this point. We have added a sentence to Strengths and Weaknesses at line 463: “However, our data series finished in March 2017 and analysis of more recent data is required, especially covering the period of the coronavirus pandemic.”

I agree with the conclusion that findings are of practically no increased risk for many of the outcomes analyzed, or at most very mild measures of increased relative risk that cannot exclude residual confounding. I wonder, given the very small relative risk at the initial period, particularly after accounting for other factors, if it would have been expected to see a reduction because the starting point is already quite similar. Authors comment about looking for potentially more sensitive outcome measures which I also find interesting.

We agree that that a reduction over time is unlikely given that group differences were small at the starting point. However, that pertains to the controlled group differences, which were not known until we conducted our analysis. Uncontrolled differences were quite large at the start hence it was reasonable to explore if group differences had reduced over time. We were rather surprised to find that the controlled differences were so small! However, we recognise that the wording of our Discussion and Conclusion may confusingly give the impression that we are looking to account for something already very small. We have therefore revised the discussion and conclusion to avoid this impression, which has mostly involved re-arranging the existing text.

---

## [Editor Report · Decision Letter 1]

16 Jan 2023

Retrospective study of more than 5 million emergency admissions to hospitals in England: epidemiology and outcomes for people with dementia

PONE-D-22-26774R1

Dear Dr. Reeves,

We’re pleased to inform you that your manuscript has been judged scientifically suitable for publication and will be formally accepted for publication once it meets all outstanding technical requirements.

Kind regards,

Sreeram V. Ramagopalan

Academic Editor

PLOS ONE
---

## [Editor Report · Acceptance letter]

27 Feb 2023

PONE-D-22-26774R1 

 Retrospective study of more than 5 million emergency admissions to hospitals in England: epidemiology and outcomes for people with dementia 

Dear Dr. Reeves:

I'm pleased to inform you that your manuscript has been deemed suitable for publication in PLOS ONE. Congratulations! Your manuscript is now with our production department. 

Kind regards, 

on behalf of

Dr. Sreeram V. Ramagopalan 

Academic Editor

PLOS ONE